# Secondary Dengue Infection Elicits Earlier Elevations in IL-6 and IL-10 Levels

**DOI:** 10.3390/ijms252011238

**Published:** 2024-10-19

**Authors:** Sonia L. Espindola, Jessica Fay, Graciela M. Carballo, Matías J. Pereson, Natalia Aloisi, María Noel Badano, Julián Ferreras, Carina Argüelles, Simón Pezzarini, Roberto Chuit, Marcos Miretti, Federico A. Di Lello, Patricia Baré

**Affiliations:** 1Laboratorio GIGA, Instituto de Biología Subtropical (IBS), Consejo Nacional de Investigaciones Científicas y Técnicas (CONICET), Facultad de Ciencias Exactas Químicas y Naturales, Universidad Nacional de Misiones (UNaM), Posadas 3300, Argentina; sonialespindola@gmail.com (S.L.E.); jessicavfay@gmail.com (J.F.); juf2003@gmail.com (J.F.); franciscarguelles@yahoo.com (C.A.); simonpezzarini@gmail.com (S.P.); mmiretti03@yahoo.co.uk (M.M.); 2Laboratorios CEBAC SRL, Posadas 3300, Argentina; gachicar@gmail.com; 3Instituto de Medicina Experimental (IMEX), Consejo Nacional de Investigaciones Científicas y Técnicas (CONICET), Academia Nacional de Medicina, Buenos Aires 1425, Argentina; matiasjpereson@gmail.com (M.J.P.); noebadano@yahoo.com.ar (M.N.B.); fadilello@gmail.com (F.A.D.); 4Instituto de Investigaciones Hematológicas (IIHEMA), Academia Nacional de Medicina, Buenos Aires 1425, Argentina; naloisi@hotmail.com; 5Instituto de Investigaciones Epidemiológicas (IIE), Academia Nacional de Medicina, Buenos Aires 1425, Argentina; rchuit@gmail.com

**Keywords:** dengue virus, cytokines, infection phases, secondary infection, febrile stage

## Abstract

This study investigates the kinetics of interleukine-6 (IL-6) and interleukine-10 (IL-10) levels in dengue virus (DENV) infections during the febrile stage. Viremic patients were categorized into two phases based on anti-DENV IgM presence. Among 259 patients, 71% were in Phase I and 29% in Phase II. Secondary infections, accounting for 38.2% of cases, exhibited earlier elevations of IL-6 and IL-10 than primary infections, suggesting that pre-existing immune memory primes faster cytokine release. Thrombocytopenia and elevated aspartate transaminase (AST) levels were associated with Phase II, secondary infections, and hospitalization. Elevated IL-6 and IL-10 levels correlated with low platelet counts, linking them to clinical manifestations. The key finding is that IL-6 and IL-10 levels rise earlier in secondary infections compared to primary infections, whereas elevated cytokine levels typically occur later in the febrile phase. This study highlights the importance of cytokine dynamics in DENV infections, particularly during the early stages. The observation of cytokine concentration changes, especially in viremic samples, provides insights into the progression of dengue disease. Further research with broader cytokine panels is warranted to validate and expand these findings.

## 1. Introduction

Dengue is a human disease caused by an infection with any one of four genetically related dengue virus serotypes (DENV-1, -2, -3 and -4), which are transmitted to individuals through the bite of infected mosquitoes belonging to some Aedes species. Multiple factors have contributed to the reemergence of DENV infections as a significant global public health concern during the past two decades [1].

DENV infection can vary in presentation, ranging from asymptomatic cases or mild illness to severe disease. Severe dengue is characterized by vascular permeability, plasma leakage, massive bleeding, and, in some cases, liver compromise, organ impairment, and even death [2].

In patients who experience a sudden deterioration of symptoms during the critical phase, which typically occurs around 2–7 days after the onset of illness, close monitoring is essential to prevent complications and reduce the risk of mortality [3].

The known contributors behind severe and fatal cases of dengue are viral serotype [4], a second heterotypic infection which could cause an antibody-dependent enhancement (ADE) [5,6], and different aspects of the host’s innate and adaptive immune response [7]. Cytokines, mediators released due to complex interactions between DENV and the host immune responses, have been implicated in the progression of severe dengue disease [8,9]. Excessive generation of pro-inflammatory cytokines, such as IL-6, has been proven to contribute to the production of antiplatelet or anti-endothelial cell antibodies, which results in a deficiency in coagulation, leading to bleeding with dengue infection [10,11,12]. Conversely, the presence of anti-inflammatory cytokines such as IL-10 leads to compromised immune clearance and persistent infectious effects during acute viral infection [13,14].

The current application of these potential markers remains uncertain due to the limited understanding of how their expression evolves during the initial phases of the infection and whether their increase is related to the disease outcome. 

Because previous reports indicated that the events triggering the release of inflammatory mediators take place very early in illness (specifically, during the initial period of the febrile phase) [15], and in line with the Centers for Disease Control and Prevention (CDC) description of the febrile phase as having a biphasic nature [3], our study introduces a classification of two sub-phases within the febrile phase. In this context, we examined the levels of IL-6 and IL-10 to analyze their kinetics in patients with primary or secondary DENV infection across these two distinct febrile phases. Our main contribution relies on the observation of cytokine concentrations and their changes at the initial period of the febrile phase, very early in disease, and their association with clinical manifestations.

## 2. Results

### 2.1. Characteristics of the Study Population

Sera or plasma samples from 259 patients referred for dengue infection diagnosis during the 2016 to 2020 period were included in this study. Consistent with the CDC description that the febrile stage is biphasic in nature, our study introduces a two-subphase classification within the febrile phase based on the presence or absence of IgM antibodies. We found 184 patients (71.0%) in Ph I (IgM negative) and 75 patients (29%) in Ph II (IgM positive). One-hundred and thirty-seven individuals (52.9%) were female. Gender showed similar distribution between phases (Table 1). The median age was 44, with quartiles being Q1–Q3: 29–63 years. Ph I patients were, on average, 10 years younger than Ph II patients [41 years (29–60) vs. 51 (29–72), respectively, *p* = 0.031]. Furthermore, as shown in Table 1, our analysis revealed a prevalence of 86.3% (*n* = 177) for DENV-1 and 13.7% (*n* = 28) for DENV-4. Since all serum samples were viremic and collected during the first week after symptom onset, the absence or presence of anti-dengue IgG antibodies was used to classify patients with primary or secondary DENV infection, respectively. Of the 259 patients, 160 (61.8%) had primary infection, while the remaining 99 patients (38.2%) exhibited evidence of secondary infection.

### 2.2. Cytokine Levels According to Febrile Phases and Type of Infection

To establish reference levels, IL-6 and IL-10 cytokines were quantified in healthy donors from blood banks (*n* = 50), along with the group of 259 DENV serum samples included in the study. The median for IL-6 concentration was 3.48 pg/mL in healthy controls and 6.71 pg/mL in DENV-infected patients (*p* < 0.0001), while for IL-10 levels, the values were 5.75 pg/mL in healthy individuals and 16.73 pg/mL for those infected with DENV (*p* < 0.0001) (Figure 1).

The analysis of cytokine levels in relation to the febrile stage and the type of infection revealed that IL-6 and IL-10 concentrations were elevated in Ph I from secondary infections compared to Ph I in primary infections (*p* = 0.005 and *p* < 0.001, respectively). Conversely, primary infections showed higher values in Ph II than secondary infections for both cytokines (IL-6, *p* = 0.009; IL-10, *p* = 0.003) (Figure 2).

### 2.3. Laboratory Findings

Analyzing clinical parameters and cytokines between the early and late phases, a significant increase in the proportion of patients with platelet count below 100,000/mL in Ph II compared to Ph I was observed (*p* = 0.007). Additionally, a higher proportion of patients in Ph II had elevated AST values compared to those in Ph I (*p* = 0.017). Table 2 presents the clinical characteristics analyzed according to the phase within the febrile stage.

Considering the observed deviations in laboratory parameters during Ph II, our final assessment involved a thorough examination of the relationship between these parameters and cytokine concentrations. Correlation analyses were conducted to examine the relationship between cytokines (IL-6 and IL-10) and biochemical parameters (platelets, white blood cell count, and liver transaminases), including individuals in the two phases and with both types of infection. Significant correlations were observed only in Ph II samples from individuals with secondary infections (*n* = 48). In summary, the analysis revealed that elevated IL-6 levels were associated with low platelet counts and tended to be associated with high ALT. On the other hand, elevated IL-10 levels were associated with WBC < 5000, low platelets, and increased AST (Table 3). Finally, regarding patients in Ph II with primary infection, only a significant association was observed for IL-6 and white blood cells (Appendix A). 

### 2.4. Characteristics of Hospitalized Patients

Hospitalization often indicates greater disease severity in dengue infection. Among our cohort, 26 patients (10%) required hospitalization due to DENV infection complications, prompting a comprehensive analysis of their clinical and immunological parameters. Our study revealed that, on average, hospitalized patients were 15 years older than outpatients (*p* = 0.007). Among them, a higher proportion exhibited platelet values below 100,000/µL compared to non-hospitalized patients (*p* < 0.001). Additionally, hospitalized patients displayed elevated IL-6 and AST concentrations, highlighting the severity of the condition (*p* < 0.001 and *p* = 0.047, respectively), (Table 4).

## 3. Discussion

IL-6 and IL-10 are two cytokines known to play critical roles in the pathogenesis of DENV infection [16,17]. Their levels can vary depending on the type of infection and may influence the course of the disease, justifying further research to better understand their significance in disease progression, especially given the persistent controversy observed [14,18]. Previous research has indicated that patients with more severe clinical disease often have higher levels of serum IL-6 and IL-10 [11,19,20]. Despite many attempts to further understand the immunoregulatory events associated with mild and hemorrhagic forms of DENV infection, the lack of a clear consensus on the timing and parameters for measuring severity prevents consistent conclusions. 

Recent studies have indicated that cytokine levels can rise in the early stages of infection, possibly triggered by the innate immune response following virus entry [21]. Nevertheless, IL-6 and IL-10 effects in the course of viral infections depend on its spatial and temporal delivery. To explore this, considering this condition, we divided the febrile viremic stage into two phases and examined the kinetics of IL-6 and IL-10 levels in a limited window period, introducing a classification of two sub-phases within the febrile phase. 

When we analyzed the values of cytokines IL-6 and IL-10 based on the type of infection (primary or secondary) and the phase of infection, we observed an interesting pattern. In secondary dengue infections, cytokine levels tended to rise earlier in the course of the disease compared to primary infections, in which elevated cytokine values were detected during the later febrile phase (Ph II). Since the pre-existing immune memory leads to an anticipated immune response, a more rapid cytokine increase is observed. This could be associated with the appearance of specific anti-dengue antibodies, both IgM and IgG, which emerge faster in secondary infections [22]. 

At the same time, secondary infections displayed lower levels of IL-6 and IL-10 during Ph II compared to primary DENV infections. This might indicate the resolution of the infection or defervescence in most of the patients. If elevated levels correlate with clinical outcomes, such as thrombocytopenia, liver damage, and the need for hospitalization, it might be possible to use certain cytokines as biomarkers to predict these complications. 

The role of cytokine levels in the immunopathogenesis of dengue is still a topic of discussion in the literature; in this regard, our work demonstrates that it is crucial to establish a limited window period (phases) to evaluate cytokine levels in primary or secondary infections.

Although reports of elevated IL-6 levels in the early acute phase in bleeding patients were published [11], others observed significantly higher IL-6 levels associated with hemorrhagic fever only in patients infected with DENV-2, but not with DENV-1 [19]. Likewise, high levels of IL-10 during early illness were an indicator of an altered antiviral response, and this association was particularly observed in patients who progressed to dengue hemorrhagic fever [20,23].

As expected, in the present work, cytokine values for healthy controls were similar to those observed for healthy populations in previous reports, but lower than the levels exhibited in the group of infected patients [24,25]. Understanding the typical range of cytokine concentrations in serum for a specific group of healthy controls can aid in identifying patients at risk of developing severe complications or predicting disease progression. 

Thrombocytopenia (platelet count < 100,000/µL) is a hallmark of dengue hemorrhagic fever, and it remains one of the current criteria for diagnosing the condition [26]. We observed a significant increase in the proportion of individuals with low platelet counts and elevated AST values in Ph II. Focusing on the very early stages of the disease, when some laboratory parameters were still within normal ranges, may have limited our ability to observe further significant clinical alterations. Additionally, patients in Ph II were older than those in Ph I, suggesting that age may have impacted disease outcomes, although this factor cannot be conclusively evaluated.

We found that about 40% of our cases were secondary infections. Secondary dengue infections are more likely to trigger severe disease even as the viral load decreases [5,6,27]. When analyzing correlations between the type of infection and different laboratory parameters in the group with secondary infection, we identified a higher incidence of individuals with platelet counts below 100,000/µL, consistent with previous reports in which the risk of severe dengue increased in patients with heterotypic secondary DENV infection [28,29]. Since correlations were only observed for the group of individuals with secondary infections in Ph II, it seems plausible that elevated IL6 and IL10 levels in Ph I might be indicative of deteriorating conditions in Ph II. As evidence for this, we should carry out a longitudinal study of the patients, which was not conducted in this work. 

Of the 259 dengue-positive samples analyzed, DENV-1 was the predominant serotype, while DENV-4 accounted for a smaller portion. The serotype distribution in the cohort is crucial, as it may influence our results. DENV-1 was the most prevalent circulating serotype until DENV-4 was introduced in Misiones in 2019, a pattern consistent with previous publications [30,31]. Hence, the proportion of DENV-4 patients in our study was small, not allowing us to perform further analyses based on serotypes. In addition, the lack of comprehensive information on all serotypes is another limitation of our study.

It is essential to note that when sequential DENV infections are closely spaced, significant cross-protection may occur, potentially modifying the association between the type of infection and the severity [32]. In this study, we were not able to distinguish between homotypic and heterotypic infections, which is an important aspect of the infection outcome [28,29].

Finally, the analysis of hospitalized patients in this cohort revealed altered AST concentrations and platelet counts combined with elevated levels of IL-6, highlighting these three parameters as warning signs. The older median age in the hospitalized group compared to outpatients may have contributed to the increased risk. However, the results of this study allow us to hypothesize that an exacerbated cytokine response, along with other altered factors such as liver dysfunction and thrombocytopenia, contributes to disease progression, as is concordant with previous reports [33]. This hypothesis is supported by the elevated IL-6 levels observed in the hospitalized patients in our cohort.

It is important to note that cytokine elevation does not necessarily correlate with a poor prognosis, as cytokines also play a vital role in pathogen eradication [34]. Therefore, for a more accurate understanding of how cytokines regulate infection control, a temporal and spatial refinement of the study is needed. This would aim to elucidate the delicate balance between inflammation and disease resolution.

### Study Limitations

While the study boasts several strengths, such as its large sample size; inclusion of patients from different outbreaks belonging to the same population in Posadas, Misiones, Argentina; and the separation of two febrile phases, it is essential to acknowledge its limitations. This is not a sequential longitudinal study; rather, samples from different phases correspond to distinct individuals, each at a specific point in their infection. Moreover, the study focused on a limited set of cytokines, and further studies should explore the role of other immune factors in dengue infection. 

## 4. Materials and Methods

### 4.1. Samples and Study Design

This is a retrospective study carried out on 259 archived plasma/serum samples collected from patients in the city of Posadas, Misiones (northeastern Argentina), referred for diagnosis of dengue infection during dengue epidemic periods between 2016 and 2020. Serum samples from DENV-infected patients were confirmed for the presence of specific NS1 DENV antigen (NS1-Ag) by immunochromatographic test (SD BIOLINE Dengue DUO kit, Abbott, Green Oaks, IL, USA) or virus genome detection (DENV-RNA) using the RealStar^®^ Dengue RT-PCR Kit 3.0 (Altona Diagnostics, Hamburg, Germany). Specific antibodies to DENV (IgM and IgG) were determined by ELISA (Dia.Pro Diagnostic Bioprobes s.r.l., Milan, Italy). DENV serotypes were determined in 205 patients using the CDC DENV 1–4 real-time PCR assay modified by Santiago et al. [35]. We retrieved patient clinical data from the Laboratory system.

### 4.2. Classification of Phases and Type of Infection 

Patients within the febrile stage were categorized into two phases: the early phase (Ph I), characterized by the presence of DENV-RNA (+) and/or NS1-Ag (+), and the late phase (Ph II), characterized by the presence of viremia along with anti-DENV IgM. The presence of specific anti-DENV IgG antibodies has been used to distinguish between primary and secondary dengue infections. Secondary infections were determined when NS1-Ag or a viral genome were present concurrently with anti-DENV IgG antibodies, and patients with viremia but without anti-DENV IgG antibodies were categorized as primary infections irrespective of the presence of anti-DENV IgM.

### 4.3. Biochemical Assays 

Routine lab determinations were assessed in the cohort. Platelet and white blood cell counts were performed with a CELL-DYN Ruby Hematology Analyzer (Abbott Diagnostic). The concentrations of hepatic enzymes, ALT (serum alanine transaminase), and AST (serum aspartate transaminase), as well as total protein, albumin, and bilirubin, were determined by photometric methods using the ALINITY CiSeries system (Abbott Diagnostic). The hepatic enzymes were determined in 210 patients, and total protein, albumin, and bilirubin were determined in 150, 166, and 150 patients, respectively. Parameters were categorized according to specific thresholds that have previously been associated with dengue severity [15,26]. White blood cell counts were considered indicative of severity when they fell below 5000/μL, and for platelet counts, below 100,000/μL. Liver enzyme concentrations exceeding the upper limit of the normal reference range (40 IU/mL) were considered elevated. 

### 4.4. Cytokine Determination in Serum Samples

The IL-6 (standard curve range: 0–300 pg/mL) and IL-10 (standard curve range: 0–500 pg/mL) concentrations were determined using commercial reagents based on enzyme-linked immunosorbent assay (ELISA) (BD-Biosciences, San Diego, CA, USA). Procedures were carried out following the supplier’s recommendations. Duplicate procedures were performed in selected samples to verify the accuracy of the results. A control group comprising 50 healthy blood donors testing negative for dengue antibodies was included to establish cytokine reference ranges. 

#### Statistical Analysis

Continuous variables were compared using either Student’s *t*-test or the Mann–Whitney U test depending on their distribution. Categorical variables were assessed using the Chi-square test or Fisher’s exact test. Confidence intervals were set at 95% (CI95), and a *p* value < 0.05 was considered statistically significant. Statistical analysis was conducted using the SPSS statistical software package 23.0 (IBM SPSS Inc., Chicago, IL, USA), and graphical representations were generated using GraphPad Prism 10.2.3 software. 

### 4.5. Ethical Aspects

The experimental protocols and procedures carried out in this work were approved by the Biosafety Review Board, the Ethical Committee of the Academia Nacional de Medicina, Buenos Aires (CEIANM) (TI 12243/16/X), and the Ethical Committee of the “Investigación Provincial”, Misiones (CEIP). Patients provided written informed consent to have their samples and data from their medical records used in research. In all cases, the confidentiality of the data was preserved, and the anonymity of the samples was maintained, in accordance with national and international standards. 

## 5. Conclusions

In summary, our study provides valuable insights into the kinetics of IL-6 and IL-10 during DENV infection. The association between these cytokines highlights the importance of categorizing patients at DENV diagnosis by different phases of the early febrile stage and type of infection, whether primary or secondary. Measuring cytokine levels as potential indicators of disease progression could be an invaluable tool for earlier patient interventions, especially in secondary infections. Future studies should be conducted to validate our conclusions and explore the relationship between a broader panel of cytokines and larger patient populations experiencing severe complications and hospitalization to determine the predictive value of cytokine levels in DENV infections. 

## Figures and Tables

**Figure 1 ijms-25-11238-f001:**
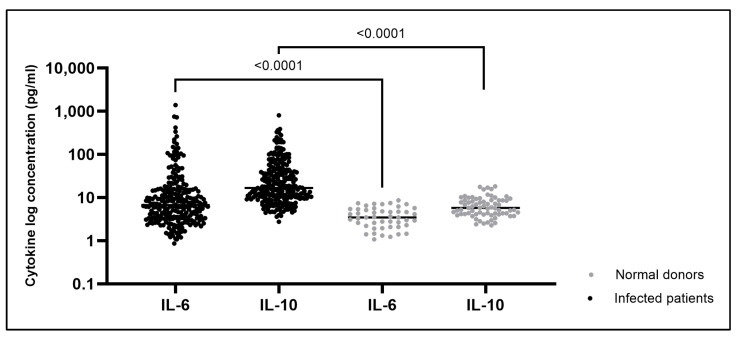
Normal ranges for Interleukin-6 and Interleukin-10 levels in uninfected individuals and in patients infected with DENV. Cytokine values in the DENV-infected group (in black) and the healthy population (in grey). Median for Interleukin-6 and Interleukin-10 concentrations are expressed in log (pg/mL). Median concentrations were compared using the Mann–Whitney U test and a *p* value < 0.05 was considered statistically significant.

**Figure 2 ijms-25-11238-f002:**
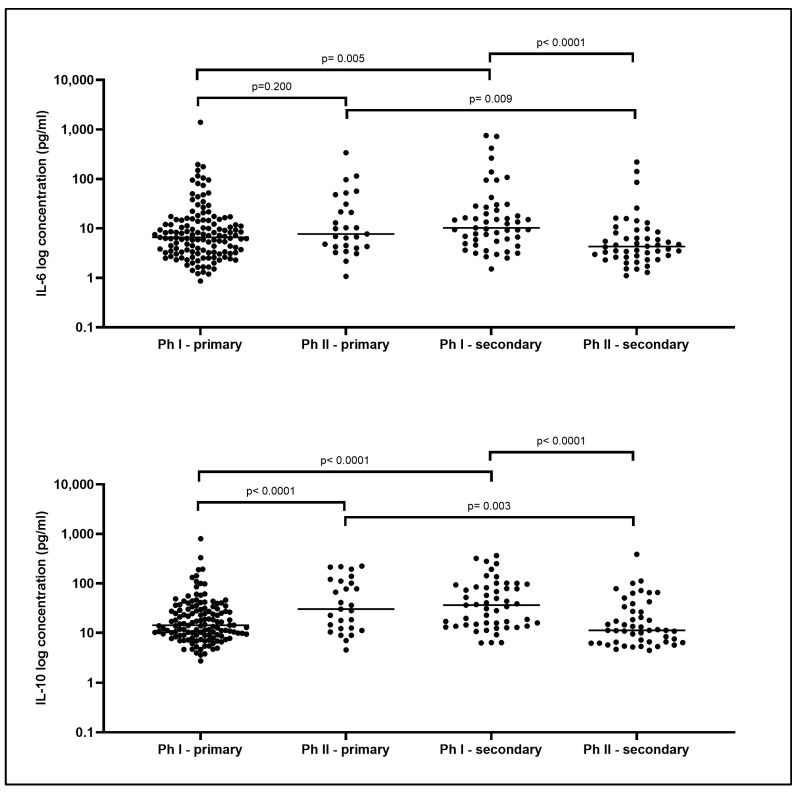
Cytokine levels in different groups. Cytokine levels at different phases and type of infection (*n* = 259). Values are expressed in log (pg/mL). Median concentrations were compared using the Mann–Whitney U test and a *p* value < 0.05 was considered statistically significant.

**Table 1 ijms-25-11238-t001:** Epidemiological and clinical characteristics by phase (*n* = 259).

Characteristics	Total (*n* = 259)	Phase I (*n* = 184)	Phase II (*n* = 75)	*p*
Median age *	44 y (29–63)	41 y (29–60)	51 y (29–72)	0.031
Female gender (%)	137 (52.9)	98 (53.3)	39 (52)	0.891
Dengue serotype (%) ^#^				
1	177 (86.3)	131 (82.9)	46 (97.9)	
4	28 (13.7)	27 (17.1)	1 (2.1)	0.007
Infection (%)				
Primary	160 (61.8)	133 (72.3)	27 (36.0)	
Secondary	99 (38.2)	51 (27.7)	48 (64.0)	<0.001

* Age in years (y) is presented as median and the interquartile range is reported in parentheses. ^#^ Serotype was available in 205 patients. Age was compared using the Mann–Whitney U test. Categorical variables were assessed using the Chi-square test or Fisher’s exact test. Confidence intervals were set at 95% (CI95) and a *p* value < 0.05 was considered statistically significant.

**Table 2 ijms-25-11238-t002:** Clinical characteristics by phase within the febrile stage.

Clinical Parameters	Total (*n* = 259)	Phase I (*n* = 184)	Phase II (*n* = 75)	*p*
WBC < 5000/µL (%)	170 (65.6)	119 (65.0)	51 (68.9)	0.663
Platelets < 100,000/µL (%)	29 (11)	14 (7.8)	15 (20.8)	0.007
AST ≥ ULN (U/L) ^#^ (%)	98 (46.7)	65 (41.7)	33 (61.1)	0.017
ALT ≥ ULN (U/L) ^#^ (%)	63 (30)	45 (28.8)	18 (33.3)	0.606
Albumin g/dL ^#^	4.1 (3.9–4.4)	4.21 (3.9–4.4)	4.04 (3.8–4.2)	0.005
Bilirubin mg/dL ^#^	0.49 (0.34–0.73)	0.48 (0.34–0.73)	0.50 (0.34–0.74)	0.391
Total protein g/dL ^#^	6.99 (6.7–7.3)	6.99 (6.7–7.4)	7.00 (6.7–7.2)	0.446

WBC: white blood cells, ULN: upper limit of normal (>40 U/L). ^#^ Available in: Alanine aminotransferase (ALT) and AST 210 patients (156 for Phase I and 54 for Phase II), albumin 160 patients, total protein, and bilirubin 150 patients. Continuous variables such as albumin, bilirubin, and total proteins were compared using the Mann–Whitney U test. Categorical variables were assessed using the Chi-square test or Fisher’s exact test. Confidence intervals were set at 95% (CI95) and a *p* value < 0.05 was considered statistically significant.

**Table 3 ijms-25-11238-t003:** Cytokine levels and clinical parameters during phase II of secondary infections (*n* = 48).

Characteristics	IL-6 *	*p*	IL-10 *	*p*
WBC 5000/µL				
<	4.3 (2.9–7.6)	0.891	14.0 (9.8–37.6)	0.014
>	3.8 (2.3–10.7)		6.6 (5.7–11.2)	
Platelets 100,000/µL				
<	10.7 (3.5–15.7)	0.020	42.3 (11.6–100.6)	0.002
>	4.2 (2.6–6.2)		10.8 (6.2–18.0)	
AST ULN (U/L) ^#^				
<	3.3 (2.0–8.5)	0.252	7.6 (5.7–13.5)	0.006
>	4.9 (3.5–10.7)		17.2 (11.3–42.3)	
ALT ULN (U/L) ^#^				
<	3.5 (2.8–6.3)	0.082	11.2 (6.3–20.5)	0.330
>	6.2 (3.5–15.2)		17.2 (8.3–30.5)	

* Median (interquartile range), WBC: white blood cells, ULN: upper limit of normal (>40 U/L). ^#^ Available in 34 patients. Median values were compared using the Mann–Whitney U test.

**Table 4 ijms-25-11238-t004:** Epidemiological and clinical characteristics of DENV-infected patients by hospitalization, *n* = 259.

Characteristics	Outpatients (*n* = 233)	Hospitalized (*n* = 26)	*p*
Age * (years)	42 (28–62)	57 (40–74)	0.007
Female gender (%)	121 (51.9)	16 (61.5)	0.235
Infection			
Primary	145 (62.2)	15 (57.7)	
Secondary	88 (37.8)	11 (42.3)	0.401
WBC < 5000/µL (%) ^#^	152 (65.8)	18 (69.2)	0.455
Platelets < 100,000/µL ^#^	19 (8.4)	10 (40)	<0.001
IL-6 pg/mL	6.21 (3.2–14.7)	11.07 (7.4–62.8)	<0.001
IL-10 pg/mL	16.07 (9.4–40.8)	21.72 (11.5–62.6)	0.235
AST ≥ ULN (U/L) ^#^	83 (44.4)	15 (65.2)	0.047
ALT ≥ ULN (U/L) ^#^	56 (29.9)	7 (30.4)	0.566

* Median (interquartile range), WBC: white blood cells, ULN: upper limit of normal (>40 U/L). ^#^ WBC available in 231 outpatients, platelets available in 226 outpatients and 25 hospitalized patients, ALT and AST available in 210 patients. Continuous variables were compared using the Mann–Whitney U test. Categorical variables were assessed using the Chi-square test or Fisher’s exact test. Confidence intervals were set at 95% (CI95) and a *p* value < 0.05 was considered statistically significant.

## Data Availability

The datasets used and/or analyzed during the current study are available from the corresponding author upon reasonable request.

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
