# Peer review of "Secondary Dengue Infection Elicits Earlier Elevations in IL-6 and IL-10 Levels"

_ijms, 2024, doi:10.3390/ijms252011238_

Round 1

Reviewer 1 Report

Comments and Suggestions for Authors

The authors collected 259 patients with dengue virus (DENV) infection. They divided them into Phase I primary and secondary infection and Phase II primary and secondary infection based on the presence or absence of anti-DENV IgM or IgG. They found that secondary DENV infection elicits earlier elevations in IL-6 and IL-10 levels. They also found that thrombocytopenia and elevated AST levels were associated with higher IL-10 levels in Phase II of secondary infections and hospitalization. Though the findings are interesting, the biological significance for clinical settings remains and requires longitudinal studies with more patients.

Major:

Based on the data in Figure 2, IL-10 levels were significantly higher in Phase II of primary DENV infection patients compared to Phase I. Secondary DENV infection patients exhibited earlier elevations in IL-6 and IL-10 levels. Still, the levels of IL-6 and IL-10 decreased to lower than those in primary DENV infection patients in Phase II. According to Table 3, higher IL-10 was associated with thrombocytopenia and increased AST, which were also linked to hospitalization. What is the biological significance of these findings in clinical settings? For example, if the IL-10 levels of secondary DENV infection patients in Phase II remain as high as in Phase I, can we predict that these patients are at risk of developing thrombocytopenia, liver damage, and requiring hospitalization? Conversely, for primary DENV infection patients, if IL-10 levels do not increase in Phase II, can we predict that these patients are less likely to develop thrombocytopenia or liver damage in the future?

Minor

1.      Please remove “.” from the title.

2.      It is recommended that the information from Table 5 be merged into Table 1, combining them into a single table.

3.      Please display the full names of CK and cc in the legend of Figure 1.

4.      The sentence “Analyzing clinical parameters and cytokines between the early and late phases, a significant increase in the number of individuals with platelet count below 100,000/ml in Ph II compared to Ph I was observed (p=0.007).” (Lines 111-113) is not correct. It should be corrected to “Analyzing clinical parameters and cytokines between the early and late phases, a significant increase in the proportion of patients with platelet count below 100,000/ml in Ph II compared to Ph I was observed (p=0.007).”.

5.      Similarly, the sentence “We observed a significant increase in the number of individuals with low platelet counts and elevated AST values in Ph II.” (lines 185-187) should be replaced with “We observed a significant increase in the proportion of individuals with low platelet counts and elevated AST values in Ph II.”.

6.       In Table 2, the calculation of the percentages of patients with AST or ALT ≥ULN (U/L) in Total, Phase I, and Phase II are inconsistent.

7.      Please show the relationship between cytokines (IL-6 and IL-10) and biochemical parameters (platelets, white blood cell count, and liver transaminases) in Ph II samples from individuals with primary infections (n=27) as supplementary data.

8.      The sentence “On the other hand, elevated IL-10 levels were associated with WBC>5000, low platelets, and increased AST (Table 3). (lines 127-128) should be corrected to “On the other hand, elevated IL-10 levels were associated with WBC<5000, low platelets, and increased AST (Table 3).”.

9.      In Table 3, it is unusual that 65 (41.7) and 4.21 (3.9-4.4) appear in the p value column.

10.  The percentages of patients with WBC<5000/mL (%) and Platelet<100,000/mL in Table 4 need to be rechecked.

11.  The sentence “Of the 259 dengue-positive samples analyzed, DENV-1 was the predominant serotype, while DENV-4 accounted for a smaller portion (30%).” (lines 202-203) should corrected to “Of the 259 dengue-positive samples analyzed, DENV-1 was the predominant serotype, while DENV-4 accounted for a smaller portion (10.8%).”.

Author Response

Comments and Suggestions for Authors

The authors collected 259 patients with dengue virus (DENV) infection. They divided them into Phase I primary and secondary infection and Phase II primary and secondary infection based on the presence or absence of anti-DENV IgM or IgG. They found that secondary DENV infection elicits earlier elevations in IL-6 and IL-10 levels. They also found that thrombocytopenia and elevated AST levels were associated with higher IL-10 levels in Phase II of secondary infections and hospitalization. Though the findings are interesting, the biological significance for clinical settings remains and requires longitudinal studies with more patients.

Major:

Based on the data in Figure 2, IL-10 levels were significantly higher in Phase II of primary DENV infection patients compared to Phase I. Secondary DENV infection patients exhibited earlier elevations in IL-6 and IL-10 levels. Still, the levels of IL-6 and IL-10 decreased to lower than those in primary DENV infection patients in Phase II.

According to Table 3, higher IL-10 was associated with thrombocytopenia and increased AST, which were also linked to hospitalization. What is the biological significance of these findings in clinical settings? According to Table 3, higher IL-10 was associated with thrombocytopenia and increased AST, which were also linked to hospitalization. What is the biological significance of these findings in clinical settings? For example, if the IL-10 levels of secondary DENV infection patients in Phase II remain as high as in Phase I, can we predict that these patients are at risk of developing thrombocytopenia, liver damage, and requiring hospitalization? Conversely, for primary DENV infection patients, if IL-10 levels do not increase in Phase II, can we predict that these patients are less likely to develop thrombocytopenia or liver damage in the future?

Author’s reply:

Thank you for pointing this out. We agree that we had not adequately expressed the biological significance of cytokine values for clinical settings in the original manuscript, and we made some modifications following your suggestions.

The study of IL-6 and IL-10 levels in patients with DENV infections offers important biological insights. While our analyses suggest potential associations, these findings should be interpreted with caution and warrant further investigation. Future studies should include a broader panel of cytokines and larger cohorts with more severe cases and hospitalizations.

According to the reviewer's suggestion, the following sentence was included in the discussion section:

1- To add the information regarding the lower cytokine levels observed in Ph II from secondary infections compared to that observed in primary infections, we included the following paragraph from line 204 to 208, page 7, and paragraph 2.

At the same time, secondary infections displayed lower levels of IL-6 and IL-10 during Ph II compared to primary DENV infections. This might be indicating the resolution of the infection or defervescence in most of the patients. If elevated levels correlate with clinical outcomes, such as thrombocytopenia, liver damage, and the need for hospitalization, it might be possible to use certain cytokines as biomarkers to predict these complications.

2- To highlight the biological significance of our findings in clinical settings, we included the following sentences in the manuscript:

from line 259 to 263, page 8, and paragraph 2.

However, the results of this study allow us to hypothesize that an exacerbated cytokine response, along with other altered factors such as liver dysfunction and thrombocytopenia, contributes to disease progression concordant with previous reports [35]. This hypothesis is supported by the elevated IL-6 levels observed in the hospitalized patients in our cohort.

In Conclusion section, page 9, from line 337 to 345, page 8.

In summary, our study provides valuable insights into the kinetics of IL-6 and IL-10 during DENV infection. The association between these cytokines highlights the importance of categorizing patients at DENV diagnosis by different phases of the early febrile stage and type of infection, whether primary or secondary. Measuring cytokine levels as potential indicators of disease progression could be an invaluable tool for earlier patient interventions, especially in secondary infections. Future studies should be conducted to validate our conclusions and explore the relationship between a broader panel of cytokines and larger patient populations experiencing severe complications and hospitalization to determine the predictive value of cytokine levels in DENV infections.

Minor

  1. Please remove “.” from the title.

Author’s reply: the “.” was removed from in the revised version.

  1. It is recommended that the information from Table 5 be merged into Table 1, combining them into a single table.

Author’s reply: Table 1 and table 5 have been merged in the revised manuscript.

  1. Please display the full names of CK and cc in the legend of Figure 1.

Author’s reply: Changes have been applied accordingly.

  1. The sentence “Analyzing clinical parameters and cytokines between the early and late phases, a significant increase in the number of individuals with platelet count below 100,000/ml in Ph II compared to Ph I was observed (p=0.007).” (Lines 111-113) is not correct. It should be corrected to “Analyzing clinical parameters and cytokines between the early and late phases, a significant increase in the proportion of patients with platelet count below 100,000/ml in Ph II compared to Ph I was observed (p=0.007).”.

Author’s reply: the text was modified accordingly. In manuscript: Page 4, Lines 128-130

  1. Similarly, the sentence “We observed a significant increase in the number of individuals with low platelet counts and elevated AST values in Ph II.” (lines 185-187) should be replaced with “We observed a significant increase in the proportion of individuals with low platelet counts and elevated AST values in Ph II.”.

Author’s reply: the text was modified accordingly. In manuscript: Page 7, Lines 226-228

  1. In Table 2, the calculation of the percentages of patients with AST or ALT ≥ULN (U/L) in Total, Phase I, and Phase II are inconsistent.

Author’s reply: Thank you for pointing this out. The footnote of the Table 2 was modified in order to clarify the percentages. In particular, AST and ALT data were available in 210 patients (156 for the phase I and 54 for phase II).

  1. Please show the relationship between cytokines (IL-6 and IL-10) and biochemical parameters (platelets, white blood cell count, and liver transaminases) in Ph II samples from individuals with primary infections (n=27) as supplementary data.

Author’s reply: the relationship between cytokines (IL-6 and IL-10) and biochemical parameters (platelets, white blood cell count, and liver transaminases) in Ph II samples from individuals with primary infections was incorporated as supplementary table.

Supplementary Table: Cytokine levels and clinical parameters during phase II of primary infections (N=27)

Characteristics

IL-6*

p

IL-10*

p

WBC 5000/ µL

<

6.4 (3.4-21.0)

0.049

30.3 (11.2-139.3)

0.873

21.9 (7.5-55.4)

34.5 (15.4-77.7)

Platelets 100,000/µL

<

53.3 (8.3-276.9)

0.070

13.5 (12.2-53.5)

0.355

6.6 (3.8-23.9)

33.2 (11.0-118.0)

AST ULN (U/L) #

<

15.7 (4.0-71.0)

0.216

93.9 (9.5-143.5)

0.364

6.6 (3.7-10.9)

20.5 (10.1-47.5)

ALT ULN (U/L) #

<

6.9 (4.3-12.9)

0.727

22.6 (10.5-110.9)

0.662

10.3 (3.5-58.6)

36.1 (10.7-142.2)

*Median (interquartile range), WBC: White blood cells, ULN: upper limit of normal (>40 U/L),

#Available in 34 patients. Median values were compared using the Mann-Whitney U test.

  1. The sentence “On the other hand, elevated IL-10 levels were associated with WBC>5000, low platelets, and increased AST (Table 3).”(lines 127-128) should be corrected to “On the other hand, elevated IL-10 levels were associated with WBC<5000, low platelets, and increased AST (Table 3).”.

Author’s reply: the text was modified accordingly. Page 5, Lines 156-157

  1. In Table 3, it is unusual that 65 (41.7) and 4.21 (3.9-4.4) appear in the p value column.

Author’s reply: Thank you for pointing this out. Table 3 was modified accordingly, and 65 (41.7) and 4.21 (3.9-4.4) were removed in the revised manuscript because these numbers were automatically incorporated, somewhat by mistake, when the paper template was built during submission.

  1. The percentages of patients with WBC<5000/mL (%) and Platelet<100,000/mL in Table 4 need to be rechecked.

Author’s reply: Thank you for noting this. The actual table 3 (previously Table 4) was modified in order to clarify the percentages. In particular, WBC<5000/ µL was available in 231 Outpatients, Platelets<100,000/µL was available in 226 Outpatients and 25 Hospitalized patients. This information is now stated in the footnote of the Table.

  1. The sentence “Of the 259 dengue-positive samples analyzed, DENV-1 was the predominant serotype, while DENV-4 accounted for a smaller portion (30%).” (lines 202-203) should corrected to “Of the 259 dengue-positive samples analyzed, DENV-1 was the predominant serotype, while DENV-4 accounted for a smaller portion (10.8%).”.

Author’s reply: Thank you for your observation. The text was modified according to your suggestion, taking into account that the dengue serotype was determined in 205 patients as stated in the material and methods section. In particular, DENV-4 was detected in 28 out of the 205 (13.7%) patients tested to determine the serotype.

“Of the 259 dengue-positive samples analyzed, DENV-1 was the predominant serotype, while DENV-4 accounted for a smaller portion (13.7%)”. Page 7, lines 244-245, last paragraph.

Reviewer 2 Report

Comments and Suggestions for Authors

The study examines IL-6 and IL-10 kinetics during the febrile stage of dengue virus infections. Patients were divided into two phases based on the presence of anti-DENV IgM. Secondary infections showed earlier cytokine elevation compared to primary infections. Elevated IL-6 and IL-10 were linked to thrombocytopenia and higher AST levels, correlating with severe clinical outcomes.

Tables in this paper are intriguing. For example, in table 1, the author should clearly indicate the median age in the table to prevent confusion; numbers such as “44,” “41,” and “51” could be misinterpreted as counts rather than ages.

Secondly, the sample size for Dengue serotypes 1 and 4 totals 205, but the status of the remaining 54 samples is unclear.

 Thirdly, the arrangement of this table is inaccurate. The second column indicates a total sample size of 259, yet the numbers do not always sum to 259, except for the infection (%). This issue is also present in other tables, which include many characters and some of these could be better represented as independent graphs.

 Lastly, the authors should clearly detail the statistical analysis method in the table legend, specifying the method used. This should apply to all figures and tables in the paper.

Author Response

Comments and Suggestions for Authors

The study examines IL-6 and IL-10 kinetics during the febrile stage of dengue virus infections. Patients were divided into two phases based on the presence of anti-DENV IgM. Secondary infections showed earlier cytokine elevation compared to primary infections. Elevated IL-6 and IL-10 were linked to thrombocytopenia and higher AST levels, correlating with severe clinical outcomes.

Tables in this paper are intriguing. For example, in table 1, the author should clearly indicate the median age in the table to prevent confusion; numbers such as “44,” “41,” and “51” could be misinterpreted as counts rather than ages.

Secondly, the sample size for Dengue serotypes 1 and 4 totals 205, but the status of the remaining 54 samples is unclear.

Thirdly, the arrangement of this table is inaccurate. The second column indicates a total sample size of 259, yet the numbers do not always sum to 259, except for the infection (%). This issue is also present in other tables, which include many characters and some of these could be better represented as independent graphs.

Author’s reply: Thank you for pointing this out. We believe that the article received by the reviewers must not have included the table footnotes where these types of issues are clarified.

For example, the footnote of Table 1 clarified that values in Age refer to the median age and interquartile ranges, as well as the number of patients in whom the dengue serotype was determined. Nonetheless, modifications were made to this table to enhance the clarity regarding the patients' ages.

The same applies to the rest of the tables where the footnotes specify the number of patients for each determination, indicating that the percentages should not be calculated with the total in those particular cases. Additionally, for clarification, this information was included in the materials and methods section.

Page 8, Line 287, paragraph 1, Materials and Methods section: DENV serotypes were determined in 205 patients using the CDC DENV 1–4 real time PCR assay modified by Santiago et al. [16]”.

Page 9, first paragraph, Lines: 305-306: “The hepatic enzymes were determined in 210 patients and total protein, albumin and bilirubin were determined in 150, 166 and 150 patients respectively.”

Lastly, the authors should clearly detail the statistical analysis method in the table legend, specifying the method used. This should apply to all figures and tables in the paper.

Author’s reply: the statistical analysis method was specified in the footnote of the revised tables and figures.

Round 2

Reviewer 1 Report

Comments and Suggestions for Authors

The authors have enhanced the revised manuscript by adding the biological significance, particularly its clinical relevance, which has greatly improved the overall quality. However, a few points must be addressed before publication, as outlined below.

Minor

1.      Please include the full names of "CK" and "cc" (as shown on the Y-axis) in the legend of Figure 1.

2.      Please remove the extra space from “Interleukin -10” in the legend of Figure 1.

3.      In Table 2, the calculation of the percentages of patients with AST or ALT ≥ULN (U/L) in the “Total” column is still inconsistent. For example, the percentage of patients with AST ≥ULN (U/L) should be 46.7% (98/210), and the percentage of patients with ALT ≥ULN (U/L) should be 30% (63/210).

4.      In Table 3, please ensure that the position and style of the "P" in the p-value columns are consistent throughout the table.

5.      In the Supplementary Table, please ensure that the formatting is consistent with Table 3, including the position of symbols such as “<” and “>”.

6.      Please include a description of the Supplementary Table in the manuscript.

7.      Ensure that the notation for sample sizes (n=X) is consistent across all tables in the manuscript. For example, in Table 3 and the Supplementary Table, the "N" is capitalized, whereas other tables use lowercase "n."

8.      It is better to modify the sentence “Of the 259 dengue-positive samples analyzed, DENV-1 was the predominant serotype, while DENV-4 accounted for a smaller portion (30%).” (lines 244-245) to “Of the 259 dengue-positive samples analyzed, DENV-1 was the predominant serotype, while DENV-4 accounted for a smaller portion.”.
